# The Relationship between Allometric Growth and the Stoichiometric Characteristics of Euhalophyte *Suaeda salsa* L. Grown in Saline–Alkali Lands: Biological Desalination Potential Prediction

**DOI:** 10.3390/plants13141954

**Published:** 2024-07-17

**Authors:** Yanyan Wang, Tongkai Guo, Qun Liu, Zhonglin Hu, Changyan Tian, Mingfang Hu, Wenxuan Mai

**Affiliations:** 1State Key Laboratory of Desert and Oasis Ecology, Xinjiang Institute of Ecology and Geography, Chinese Academy of Sciences, Urumqi 830011, China; wangyanyan211@mails.ucas.ac.cn (Y.W.); liuqun21@mails.ucas.ac.cn (Q.L.); tianchy@ms.xjb.ac.cn (C.T.); hmf@ms.xjb.ac.cn (M.H.); 2University of Chinese Academy of Sciences, Beijing 100049, China; 3College of Water Resources and Civil Engineering, China Agricultural University, Beijing 100083, China; guotongkai@cau.edu.cn; 4Department of Production and Operation, Xinjiang Oilfield Company, Petrochina, Karamay 834000, China; huzhonglin@petrochina.com.cn

**Keywords:** euhalophyte, allometric analysis, morphological adjustment, stoichiometric characteristics, biological desalination, saline soil

## Abstract

The morphological adjustments of euhalophytes are well-known to be influenced by the soil-soluble salt variation; however, whether and how these changes in morphological traits alter the biomass allocation pattern remains unclear, especially under different NaCl levels. Therefore, an allometric analysis was applied to investigate the biomass allocation pattern and morphological plasticity, and the carbon (C), nitrogen (N), and phosphorus (P) stoichiometric characteristics of the euhalophyte *Suaeda Salsa* (*S. salsa*) at the four soil-soluble salt levels of no salt (NS), light salt (LS), moderate salt (MS), and heavy salt (HS). The results showed that soil-soluble salts significantly change the biomass allocation to the stems and leaves (*p* < 0.05). With the growth of *S. salsa*, the NS treatment produced a downward leaf mass ratio (LMR) and upward stem mass ratio (SMR); this finding was completely different from that for the salt treatments. When *S. salsa* was harvested on the 100th day, the HS treatment had the highest LMR (61%) and the lowest SMR (31%), while the NS treatment was the opposite, with an LMR of 44% and an SMR of 50%. Meanwhile, the soil-soluble salt reshaped the morphological characteristics of *S. salsa* (e.g., root length, plant height, basal stem diameter, and leaf succulence). Combined with the stoichiometric characteristics, N uptake restriction under salt stress is a vital reason for inhibited stem growth. Although the NS treatment had the highest biomass (48.65 g root box^−1^), the LS treatment had the highest salt absorption (3.73 g root box^−1^). In conclusion, *S. salsa* can change its biomass allocation pattern through morphological adjustments to adapt to different saline–alkali habitats. Moreover, it has an optimal biological desalting effect in lightly saline soil dominated by NaCl.

## 1. Introduction

Soil salinization continues to be aggravated in the arid and semi-arid regions where soil-soluble salts are abundant and various, due to insufficient rainfall for salt leaching and the excessive evaporation of soil-available water [1,2]. In these areas, the widespread distribution of saline soils has severely restricted agricultural production, but, if adequately developed and utilized, they could serve as alternative, arable land resources with which to cope with food shortages and ensure food security [3]. Currently, the remediation approaches for saline soils are no longer confined to traditional and single engineering or agronomic measures, but are paying more attention to their combination with sustainable phytoremediation [3,4]. Halophytes, especially euhalophytes, as a pioneer species colonizing natural saline habitats, can not only survive and reproduce under a NaCl concentration of approximately 0.50% or higher [5], but can also hyperaccumulate salt in their succulent organs, which is of great ecological efficacy for comprehending plant salt tolerance and developing saline agriculture. Previous studies have primarily concentrated on their morphological and physiological characteristics in response to salt stress [5,6,7,8], and the capacity underlying phytodesalination and remediation in saline soils [1,4,9,10,11]. Nevertheless, few studies have been conducted on the biomass allocation pattern of euhalophytes’ vegetative organs based on nutrient assimilation and salt acquisition. Therefore, it is essential to determine how the biomass of each organ (root, stem, and leaf) is distributed synergistically for assessing the potential of euhalophytes to acclimate to the soil-soluble salt variations in saline soils. 

Biomass allocation is mainly affected by genetic characteristics, environmental conditions, and plant size [12], and the allocation patterns obviously differ among the different species [12,13]. In accordance with optimal partitioning theory (OPT), plants are generally prone to allocate their biomass to the organs that have the easiest access to the most limited resources [14]. However, compared with glycophytes, soil-soluble salt is the primary driving factor leading to the morphological plasticity of euhalophytes under an adequate supply of nitrogen (N) and phosphorus (P) in saline soil [15,16]. Sodium (Na) is regarded as an essential element for inducing succulence formation in euhalophytes [17]. Once subjected to salt stress, euhalophytes target the transport of more nutrients to the above-ground parts for storing more salts [18]. For instance, when euhalophytes are cultured in nutrient solutions containing NaCl, leaf succulence is gradually developed through increasing the size of the epidermal cells, which helps the plants to expand the cell space to enclose more salts in the vacuoles [19]. Additionally, the root spatial distribution could directly reflect the efficiency of euhalophytes’ access to soil nutrients, water, and salts. Studies have shown that the photosynthetic products of euhalophytes are first provided for the rapid growth of roots at the seedling stage, because root elongation could ensure their acquisition of soil nutrients and their adaptation to soil-soluble salt variations [20,21]. However, the root/shoot ratio decreases significantly due to canopy increases and root turnover at a later growth stage [22], indicating that the biomass allocation during the different growth periods changes significantly, probably due to their genetic traits or morphological plasticity to environmental stress [12,23]. 

An allometric analysis could determine the quantitative relationship between a plant’s size and its apparent characteristics by analyzing its morphological plasticity [24,25]. Therein, individual size is related to the effective distribution of nutrients by plants in response to environmental stress [26,27], which significantly affects the biomass allocation proportion in the different vegetative organs [28]. Especially for N and P nutrients, their proportion in the different vegetative organs varies markedly in response to different environmental stresses, and changes in different growth periods [29]; the N/P ratio is also used to indicate the limiting conditions of plant growth due to different environmental stresses [30]. Until now, allometric analysis has rarely been used to research the specific patterns of halophytic morphological adjustment to soil salinity variations. Although Na is a key inducer of morphological changes in euhalophytes [19], it has yet to be determined how soil-soluble salts regulate their growth process and biomass allocation. Whereas, we speculate that the continuous morphological adjustments of euhalophytes in response to salinity variations may result from allometric growth. Meanwhile, we need to be clearer about the relationship between allometric patterns and stoichiometric characteristics under different soil-soluble salt levels. Euhalophyte *Suaeda salsa* (*S. salsa*) is a model vegetation widely distributed in saline habitats, and is widely used to remediate saline soils. In this study, based on two hypotheses (1) *S. salsa* relies on morphological adjustments (roots, stems, and leaves) to acclimate to soil-soluble salt variations, and (2) changes in biomass allocation pattern are an important adaptive strategy for *S. salsa* in saline soils—we conducted a trial in a root box to investigate the changes in its Na accumulation, allometry, and stoichiometric characteristics of carbon (C), N, and P under four NaCl levels to answer the questions mentioned above.

## 2. Results

### 2.1. Soil Salt and Nutrient Variations

The soil quality deteriorated to varying degrees after NaCl supplementation (Figure 1). The EC and TSS reflect the significant differences in the salinized habitats of *S. salsa* (Figure 1F,G). The lowest TSS concentration was approximately 1.64 g kg^−^^1^, found after the NS treatment, and the highest TSS was approximately about 13.44 g kg^−^^1^, found after the HS treatment (Figure 1G). *S. salsa* growing in these different salt levels led to marked changes in nutrient depletion, especially NH_4_^+^-N and NO_3_^−^-N (Figure 1A–D). The HS treatment increased the soil residual available N concentration, whereas the NS and LS had the opposite effect. With the growth progress of *S. salsa*, the NS, LS, and MS treatments significantly decreased the soil NO_3_^−^-N content (*p* < 0.05). The soil Olsen-P concentration for the NS level was higher than that for the other treatments. However, the available K in the soil slightly increased at the HS level, but significantly increased at the MS and HS levels on the 80th day.

### 2.2. Biomass Allocation

TSS, as the main restriction factor for the normal growth of *S. salsa*, could affect its proportion of biomass allocation and its allometric patterns (Figure 2 and Figure 3). The proportion of biomass allocation in the different organs was in the order of leaves > stems > roots, except for the NS treatment on sampling days 80 and 100 (Figure 2). The addition of NaCl had a significant effect on the LMR and SMR. The LMR increased significantly with an increased salt concentration, while the SMR decreased significantly. The LMR (SMR) for the HS treatment was significantly higher (lower) than that for the other treatments. In contrast, the RMR did not change significantly (Figure 2A–C). Combined with an allometric analysis of plant size, it was found that *S. salsa* at the NS level had a downward LMR with an increased total biomass (slope = −0.08857, *p* < 0.05), while the LMR for the other treatments sustained growth. Meanwhile, sharp intercept shifts showed that the LMR under the different levels was in the order of HS > MS > LS > NS (Figure 3A; *p* < 0.01). Conversely, the SMR increased significantly at the NS level for a given plant size (slope = 0.08963, *p* < 0.05). Significant intercept variations indicated that the SMR at the different levels was in the order of NS > LS > MS > HS (Figure 3B; *p* < 0.01). In contrast, the RMR slightly decreased with increased plant size, and no significant differences existed among the treatments (Figure 3C). 

### 2.3. The Morphological Traits of S. salsa 

The morphological characteristics of *S. salsa* were mostly affected by soil salinity (Figure 4). During the whole growth period, the plant height and basal stem diameter had similar response curves to the different soil salt concentrations in the order of NS > LS > MS > HS (Figure 3A,B). The leaf succulent degree experienced a declining trend with increased salt concentration, and for the NS treatment, it was the highest during the pre-growth period but the lowest during the post-growth period (Figure 4C). As growth progressed, the R/S ratio at the different sampling stages showed a curvilinear decline, particularly on the 80th day, when the R/S ratio under the four NaCl levels decreased significantly, which may have been caused by the salt accumulation traits of *S. salsa*, contributing to the above-ground biomass. However, the R/S ratio increased on the 100th day, but the differences were insignificant compared to the previous period. Furthermore, the R/S ratio for the HS treatment was higher than that for the other treatments (Figure 4D). The RL for the LS and MS treatments was longer than that for the NS and HS treatments (Figure 4E). The RA for the LS treatment had a more stable performance than that for the other treatments (Figure 4F). The SRL and SRA could reflect the ability of the roots to acquire water and nutrients, and under the different saline concentrations, the MS treatment was preferable to the other treatments (Figure 4G,H).

### 2.4. Allometric Growth Response to the Soil Salt Content

The soil salinity could significantly change the apparent morphology of *S. salsa*. Therefore, an allometric analysis was utilized to reveal the differences between the morphological traits and plant size under the four soil NaCl levels (Figure 5). Overall, variations in soil salinity significantly affected the quantitative changes in plant morphology, which could be ascribed to developmental delay, resulting in the least biomass with the HS treatment, followed by the MS treatment (Figure 5; significant shifts along the common slope, *p* < 0.01). Additionally, for the roots, the slopes of the root biomass, RL, and RA against the total biomass experienced no significant changes, respectively (Figure 5A–C, common slope, *p* > 0.05). However, the intercept shifts of the RL showed significant differences between the salt and non-salt treatments (Figure 5B; *p* < 0.01). For the stems, the slope of the stem biomass against the total biomass at the NS level was significantly higher than that at the other salt levels (Figure 5D; *p* < 0.01). Moreover, the stem biomass accumulation for the LS treatment was higher than for the MS and HS treatments (shifts in elevation, *p* < 0.01). The plant height against plant size had a similar performance as the stem biomass, and the intercept shifts (*p* < 0.01) showed that it was higher at the NS and LS levels than at the MS and HS levels (Figure 5E). The basal stem diameter against plant size for the NS and LS treatments was significantly higher than that for the MS and HS treatments (Figure 5F; different slopes were divided into two groups, *p* < 0.05). For the leaves, the slope of the leaf biomass against plant size for the NS treatment was significantly lower than that for the other treatments (*p* < 0.01), and the relationship between leaf biomass allocation and total biomass was markedly affected by soil salinity (Figure 5G). Meanwhile, the leaf succulent degree slopes were negative, and the leaf succulence decreased with an increase in plant size. For a given plant size, the leaf succulence for the NS and LS treatments was significantly higher than that for the MS and HS treatments (Figure 5H; significant intercept shifts, *p* < 0.01). 

### 2.5. Stoichiometric Characteristics of S. salsa

Throughout the entire sampling period, the different organs showed variable C concentrations, which reached optimal levels on the 100th day in the order of stems > roots > leaves. A comparison of the C concentration among the four NaCl levels showed an order of NS > LS > MS > HS, with the NS treatment being significantly higher than that for the other treatments (*p* < 0.01; Figure 6A). The N concentration of the different organs at the three sampling times was relatively constant, showing an order of leaves > stems > roots. Additionally, with an increase in the soil salt level, the roots N concentration decreased, while the N concentration in the stems and leaves increased. On the 100th day, the stems N concentration for the NS treatment was significantly higher than for the other treatments (*p* < 0.01; Figure 6B). Moreover, with an increase in the soil salt content, the phosphorus (P) concentration in each organ showed an upward trend, and the leaves P concentration was the highest amongst all the organs (Figure 6C). Given the C, N, and P concentrations in each organ, the C/P and N/P ratios presented a downward trend on day 100 (Figure 6D–F). In particular, the stems N/P ratio significantly decreased with an increase in the soil salt content, and it reached its maximum at the NS level (Figure 6F). 

### 2.6. Biological Desalination Potential Prediction

The biological desalination of *S. salsa* was focused on its adaptation to the saline soil, biomass responses, and above-ground Na content (Figure 7). The Na concentration in the above-ground vegetative organs increased significantly with an increase in the soil NaCl level (Figure 7A,B). Given the known stem and leaf biomass (Figure 7C,D), the biological desalination of *S. salsa* could be calculated by multiplying its biomass and Na concentration (Figure 7E). The amount of Na removed by *S. salsa* was the highest on the 100th day, in the order of LS > MS > HS > NS. Among them, the desalting capacity of the LS, MS, HS, and NS treatments were 3.79, 3.19, 2.43, and 1.91 g root box^−1^, respectively (Figure 7E). Although the NS had a higher biomass than that of the other treatments, it had the lowest phytodesalination due to its low Na accumulation. Thus, to predict the desalting potential of euhalophytes, it was necessary to consider their growth responses to saline habitats and their ability to accumulate salt. By comparing the desalting amount among the treatments (Figure 7E), we found that the desalting effect of *S. salsa* was optimal in the lightly saline soil (addition of 0.50% NaCl).

## 3. Discussion

### 3.1. Biomass Allocation Pattern

Plants anchored in soil need to confront a variety of biotic and abiotic challenges throughout their lifetimes; in order to survive, they must evolve their matched adaptative mechanisms to resist the interference of environmental pressures [31]. Herein, biomass allocation is the most intuitive response of plants with which to allocate limited resources to different organs to fulfill their structural and physiological functions, and to adapt to changes in the external environment [32,33]. For euhalophytes with leaf succulence, the vacuoles in the leaves can store salt. When the salt concentration is at an optimal level, it primarily supplies the normal growth of the plant, but when there is too much salt, the vacuoles have the function of diluting the salt to avoid ionic toxicity [34]. In our study, the increase in soil-soluble salts led to a “trade-off” relationship between the biomass allocation of the stems and leaves of *S. salsa*. As the growth period progressed, the “trade-off” relationship became more apparent, manifesting as a significant increase in the leaf mass ratio and a decrease in the stem mass ratio, which indicated that the incremental increases in soil-soluble salts contributed to the biomass allocation to the leaves. Previous studies have shown that halophytic biomass shows a “curvilinear” pattern in response to increased salinity in solution media, and only an optimal salinity can achieve the highest biomass [19,34]. However, our study found that euhalophytes growing in non-saline soil had the highest biomass, which indicated that the soil provided a complex medium environment for the normal growth of *S. salsa*. Similarly, this phenomenon has also been mentioned in the study of the growth response of other euhalophytes, such as *Suaeda glauca* [35], *Inula crithmoide,* and *Plantago crassifolia* [36], to increased salts. Subsequently, as the soil-soluble salt concentration increases, salt plays a dominant role in plant size. Meanwhile, the individual size significantly affects the proportion of the plant biomass allocated to the vegetative organs [26]. The larger individuals of *S. salsa* allocate more biomass to the stems and less biomass to the leaves than do smaller individuals. Meanwhile, this study also found that the root mass ratio was not significantly changed with an increase in soluble salts throughout the growth period of *S. salsa*; that is, the root biomass allocation was stable, and the R/S ratio was generally larger in high-salt soil than in low-salt soil, which would be an important adaptive strategy for tolerating salt stress. Therefore, under conditions with sufficient N and P nutrients, soil-soluble salts can mediate the biomass distribution pattern of the vegetative organs of euhalophytes by altering their plant size and morphological characteristics. 

### 3.2. Morphological Plasticity of S. salsa in Response to Soil-Soluble Salt 

Euhalophytes can accumulate considerable amounts of Na^+^ in their above-ground parts [34], and Na has always been considered an essential element for the normal growth of halophytes [37]. Although the growth curve of *S. salsa* in response to soluble salts in the soil matrix is not completely consistent with their growth performance in a nutrient solution with a series of NaCl concentrations [19,34], these changes only occur in non-saline soils with a soluble salt content of less than 0.20%. Regardless of the growth medium, the effect of Na on the growth and development of *S. salsa* is basically consistent with an increase in the salt concentration [5,38]. Na^+^ effects on the morphological characteristics of *S. salsa* have mainly been focused on leaf succulence [19,39] and root configuration [5,38]. Leaf succulence can help *S. salsa* accumulate more Na^+^ in the vacuoles, thereby reducing the toxicity caused by excess salt in the form of ion compartmentalization [38]. This also explains why high-salt treatment can significantly increase the leaf mass ratio. Additionally, based on the “trade-off “relationship between stem and leaf biomass allocation, stem thinning and shortening are the morphological adaptations to salt stress. Although the roots have a steady biomass allocation, the root morphology, e.g., the root length, root area, specific root length, and specific root area, change obviously with an increase in the soil-soluble salt. The root morphology of *S. salsa* in response to salt stress can directly reflect its ability to acquire and utilize nutrients and salts in saline soil [5,40]. The optimal performance of the root length, specific root length, and specific root area in moderately saline soil also explains the adaptive strategy of euhalophytes to soil salinity, which adjusts with increases or reductions in the root system to access salts and nutrients [5]. To sum up, in response to increased soil-soluble salts, most of the morphological traits show a “curvilinear” change during the whole growth and development process, because an appropriate amount of soil-soluble salt can promote the growth of roots, but once it becomes excessive, it will inhibit root growth [41]. Meanwhile, the allometric analysis indirectly confirms that the responses of the roots, stems, and leaves to non-saline soil can be generalized as real plasticity, yet the other morphological adjustments predict apparent plasticity. *S. salsa* can acclimate to soil salinity variations through two forms of morphological plasticity [13] to ensure their own fitness, and the apparent plasticity is particularly important.

### 3.3. Relationship between Allometric Growth and Stoichiometric Characteristics

The correlation between the allometric growth and stoichiometric characteristics of halophytes remains unclear. In our study, the allometric analysis showed that the apparent plasticity played a more important role than the real plasticity in response to salt stress. These changes in the biomass allocation and morphological traits of *S. salsa* with an increased salt concentration may be related to its stoichiometric characteristics, which can reflect the nutrient interactions between plants and their habitats [42,43]. Generally, the concentration differences in C, N, and P in each organ can determine their physiological and ecological functions [44], because the normal growth of plants depends on nutrient acquisition by the roots and nutrient assimilation by the above-ground organs [45]. In saline soils, N and P are usually deficient [1], so their supplementation is particularly important to ensure the high biomass of halophytes, because adequate N plays an important role in nutrition and osmotic regulation in euhalophytes under high salinity [46]. Koerselman and Meuleman reported that the N/P ratio could predict plant N or P limitations [47]. N was considered as a limiting factor for N/P values < 14: 1. In this study, the N/P ratio of *S. salsa* was much lower than 14, apart from that of the stems in the latter growth period, which was close to 14 in the non-saline soil, implying that the normal growth of *S. salsa* is mostly restricted by N. In other words, soil salt stress restricts N uptake by regulating the plant size of *S. salsa*. Although the N/P ratio suggests N limitation for the normal growth of *S. salsa*, the N concentration in the leaves was consistently higher than that in the other organs, and the N increased with incremental salt supplementation, indicating that the N contribution to the leaf biomass helped *S. salsa* store more salt with which to resist salt stress. Meanwhile, a high P concentration is also conducive to a greater investment in leaf biomass [48]. Based on this, leaves can easily adjust their morphology to adapt to soil salinity variations. Additionally, the C/N ratio and C/P ratio can reflect the ability of plant to use mineral nutrients to assimilate carbon [49]. We found the anabolism of *S. salsa* decreased significantly with an increase in soil-soluble salts. Consequently, biomass allocation based on the stoichiometric characteristics of *S. salsa* seem to be closely related to the allometric growth, and studying their relationship could comprehensively answer how euhalophytes acclimate to soil salinity variations through morphological adjustments, which is a synthesis of the previous research results [1,5,19]. 

### 3.4. Salt Removal Capacity of S. salsa

Biological desalination is a technique that uses halophyte restoration for saline soil [50,51,52]. This technique has been widely used in coastal, semi-arid, and arid regions, where intercropping systems between halophytes and conventional crops [9,52] or continuous cropping systems of halophytes [4,53] have been gradually established as an important technical model with which to reduce the negative impact of soil salinity on food productivity. The amount of salt removal in halophytes is determined by the salt concentration in the above-ground organs and biomass [50]. Based on this, the above-ground salt accumulation was the highest for the LS treatment, indicating that *S. salsa* grown in low salinity (0.50% NaCl addition), despite not having the highest biomass, removed more salt from the soil. Meanwhile, the allometric analysis also found that the biomass accumulation and allocation upon the LS treatment were more suitable for salt transfer to the plants to achieve phytodesalination, which is consistent with previous findings [5]. Although we obtained the desalination amount collected from the pot experiments under controlled conditions, this value does not reflect the actual situation in the field, because (i) sufficient exogenous N and P nutrients were applied to the soil for the normal growth of *S. salsa*; and (ii) the type and concentration of soil-soluble salt should be considered, rather than being limited to a NaCl treatment. Additionally, biological desalination is also related to harvest time, and a harvesting time before the leaves fall off may be the best choice [50]. Therefore, these results just provide some references for future research. 

## 4. Materials and Methods

### 4.1. Materials

Seeds of *S. salsa* were collected from the Halophyte Botanical Garden in Karamay (84°59′41.61″ E, 45°28′6.38″ N), Xinjiang Province, China. The topsoil of a cotton field cultivated at the Fukang Desert Ecosystem Station (87°45′~88°05′ E, 43°45′~44°30′ N), Chinese Academy of Sciences, was gathered for the root box trial. The soil’s initial physical and chemical properties were measured, as indicated in Table 1. A root box without drainage made from an acrylic plate (40 cm × 10 cm × 40 cm) was used to observe the root system of *S. salsa* (Figure 8A,B).

### 4.2. Experimental Design

The greenhouse experiments began on 17 June 2022, in Changji, Xinjiang Province, China (87°04′56″ E, 44°09′59″ N). Based on the actual soil salinity in the study area, four levels of NaCl were used in the experimental treatment design: no salt (NS, no added NaCl), light salt (LS, addition of 0.50% NaCl), moderate salt (MS, addition of 1.00% NaCl), and heavy salt (HS, addition of 1.50% NaCl), which simulate non-saline soil, lightly saline soil, moderately saline soil, and heavily saline soil, respectively. Every treatment was repeated 15 times. Each root box was filled with 18.00 kg of air-dried soil. To supply adequate nutrients for plant growth, fertilizers (9.54 g CO(NH_2_)_2_ and 6.90 g KH_2_PO_4_), as the basal nutrients, were dissolved in 3.80 L of deionized water and applied to the root box. Meanwhile, the soil water content was approximately 80% of the field water capacity. After 24 h, 20 intact seeds were sowed evenly in the root box and covered with air-dried soil. When the height of the seedlings was approximately 6 cm, 12 seedlings were kept per root box. Different NaCl levels (0, 90, 180, and 270 g) were dissolved in deionized water and irrigated into the root box every 3 days for a total of 5 times. During plant growth, the weighing method was applied to replenish the deionized water to field water capacity (20%, *w*/*w*).

### 4.3. Samples Collection and Analysis

Plant and soil samples, per five root boxes as a sampling unit, were collected on days 60, 80, and 100 (Figure 8C–E). After measuring the plant height and basal stem diameter, the leaves and stems were parted to measure their fresh weight. Then, they were put into an oven at 105 °C for 30 min and dried at 75 °C for 48 h to a constant weight. The soil in the root box was divided into four layers: 0–5, 5–15, 15–25, and 25–35 cm (Figure 8B). The roots in each layer were collected and washed with deionized water. The root system was scanned using a Microtek color platform scanner (Phantom9980XL, resolution 800 dpi). The images acquired were analyzed using FG-RIAS root analysis software to acquire the root length (RL) and root area (RA). The specific root length (SRL) and specific root area (SRA) were calculated by dividing the total root length and total root area by the root mass, respectively. Subsequently, the roots were placed in an oven and dried at 65 °C to a constant weight. The total biomass was calculated as the sum of the roots, stems, and leaves. The root mass ratio (RMR), stem mass ratio (SMR), and leaf mass ratio (LMR) were their proportions of the total biomass, respectively. The R/S ratio was regarded as the below-ground to above-ground biomass ratio. The degree of leaf succulence was calculated by dividing the leaf fresh weight by the leaf dry weight. The data acquired were used for the allometric analysis.

The soil samples were air-dried, ground, and sieved by 2 mm and 1 mm, respectively. The pH and electrical conductivity (EC) were measured using water extracts (pH, 1:2.5 *w*/*v*; EC, 1:1 *w*/*v*) after suspension for 30 min using a pH meter (S20, Mettler Toledo, Zurich, Switzerland) and an electrical conductivity meter (DDSJ-308A, Xi’an, China). The total soluble salt (TSS) was measured using the residue drying method. The soil-available nitrogen (N) was measured using an auto-analyzer (BRAN LUEBBE AA3, Hamburg, Germany). The phosphorus (Olsen-P) was measured using colorimetric analyses. The available potassium (K) was measured using an atomic absorption spectrometer (Thermos Electron Corporation, MA, USA).

The plant samples (roots, stems, and leaves) were crushed with a ball mill and digested with H_2_SO_4_-H_2_O_2_, and measured to determine the nitrogen (N) concentration using an Automatic Kjeldahl Analyzer (Foss 8400, Hilleroed, Denmark). The phosphorus (P) concentration was measured via colorimetric analysis, and the carbon (C) concentration was pyrolyzed and measured using a total organic carbon analyzer (Aurora Model 1030, Texas City, TX, USA). The sodium (Na) was microwave-digested with HNO_3_-H_2_O_2_ and measured using an ICP-OES (735E, Angilent, Santa Clara, CA, USA).

### 4.4. Statistical Analysis

The mean of the soil’s physical and chemical properties in the stratified sampling were the weighted averages. A one-way analysis of variance (ANOVA) followed by the least significant difference (LSD) test was applied to analyze the differences in the soil’s physical and chemical properties; its biomass, morphological traits, and stoichiometric (C, N, and P in the roots, stems, and leaves) characteristics; and the Na concentration and removal amount for the different sampling periods. All the statistical analyses were performed using Microsoft Excel and SPSS ver. 29.0 software (IBM Corp., Armonk, NY, USA).

The allometric growth of the morphological traits is generally reflected by exponential regressions shown by the equation y = β∙χ^α^, which is the logarithmic transformed into logy = β + α∙logx [25,54], where y and x are the trait variables, α is the slope, and β is the intercept. The morphological data of *S. Salsa* were log-transformed and visualized via linear analysis before the allometric analysis. The slope and intercept were acquired by SMATR (Standardized Major Axis Tests and Routines) ver. 2.0 software [55], thereby identifying specific allometric patterns between the total biomass and morphological traits under different NaCl levels. The slope differences under the different NaCl treatments (*p* < 0.05) reflected group variations. Significance tests of elevation shifts (intercept) and shifts along the common slopes were then conducted using pairwise post hoc multiple comparisons (Wald test) to determine the variations in the morphological characteristics for a given biomass under the different NaCl levels. Ultimately, an allometric analysis was performed on pairwise morphological traits to study their growth relationships under the different NaCl treatments.

## 5. Conclusions

It was found that *S. salsa* can acclimate to soil-soluble salt variations by adjusting its morphological characteristics. These morphological adjustments changed their biomass allocation pattern; for example, the LMR increased and the SRM decreased with an increase in soil salt. These morphological adjustments were further confirmed by analyzing the relationship between the allometric growth and stoichiometric characteristics. Meanwhile, the biological desalination was predicted based on the biomass allocation pattern. Since this study only considered the effects of NaCl on the growth and development of *S. salsa*, future trials should emphasize their morphological responses to diverse saline–alkali types and explore the influence of the root spatial distribution on accelerating soil salt leaching, so as to predict the biological and ecological functions of *S. salsa* in the desalination and remediation of saline–alkali habitats.

## Figures and Tables

**Figure 1 plants-13-01954-f001:**
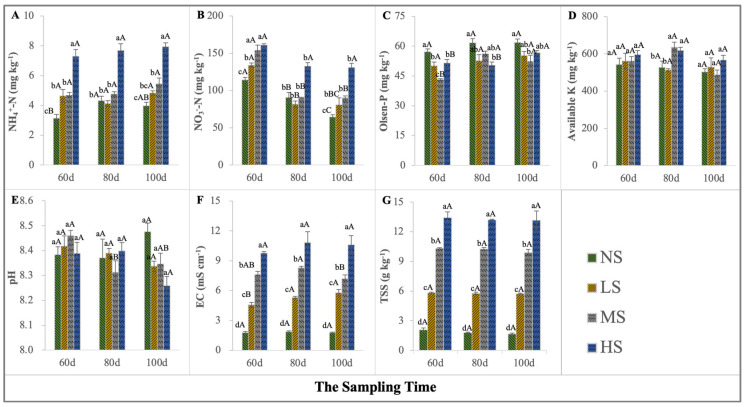
Changes in soil environmental factors under the four NaCl levels. NH_4_^+^-N (**A**), NO_3_^−^-N (**B**), Olsen-P (**C**), available K (**D**), pH (**E**), EC (**F**), and TSS (**G**) (different lowercase letters indicate significant differences among the treatments (*p* < 0.05); different uppercase letters indicate significant differences among the sampling times (*p* < 0.05)).

**Figure 2 plants-13-01954-f002:**
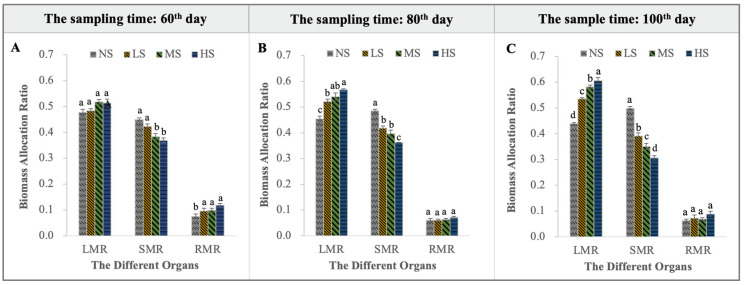
LMR, SMR, and RMR under the four NaCl levels on sampling days 60 (**A**), 80 (**B**), and 100 (**C**) (different lowercase letters indicate significant differences among the treatments (*p* < 0.05)).

**Figure 3 plants-13-01954-f003:**
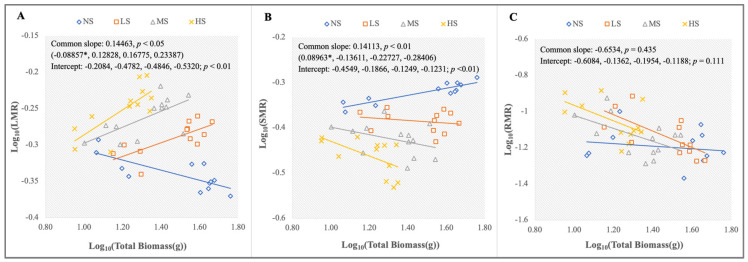
The allometric analysis of the log total biomass and the log LMR (**A**), log SMR (**B**), and log RMR (**C**) under the four NaCl levels, respectively. The common slope test (*p* < 0.05) indicates significant differences among the groups, and the group-caused differences are denoted with an asterisk. Meanwhile, (**A**,**B**) show significant shifts in elevation among the LS, MS, and HS, respectively, *p* < 0.01; (**C**) shows that the HS treatment caused a significant shift along the common slope in comparison with the other treatments (*p* < 0.01).

**Figure 4 plants-13-01954-f004:**
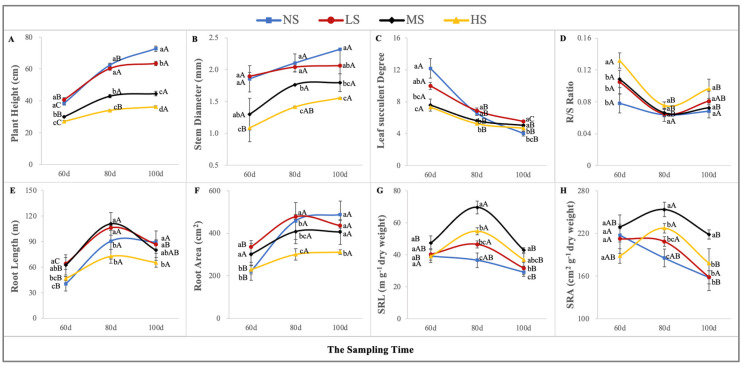
The variations in the morphological traits under the four NaCl levels throughout the sampling period. Plant height (**A**), stem diameter (**B**), leaf succulent degree (**C**), R/S ratio (**D**), root length (**E**), root area (**F**), SRL (**G**), and SRA (**H**). (Different lowercase letters indicate significant differences among the treatments (*p* < 0.05); different uppercase letters indicate significant differences among the sampling times (*p* < 0.05)).

**Figure 5 plants-13-01954-f005:**
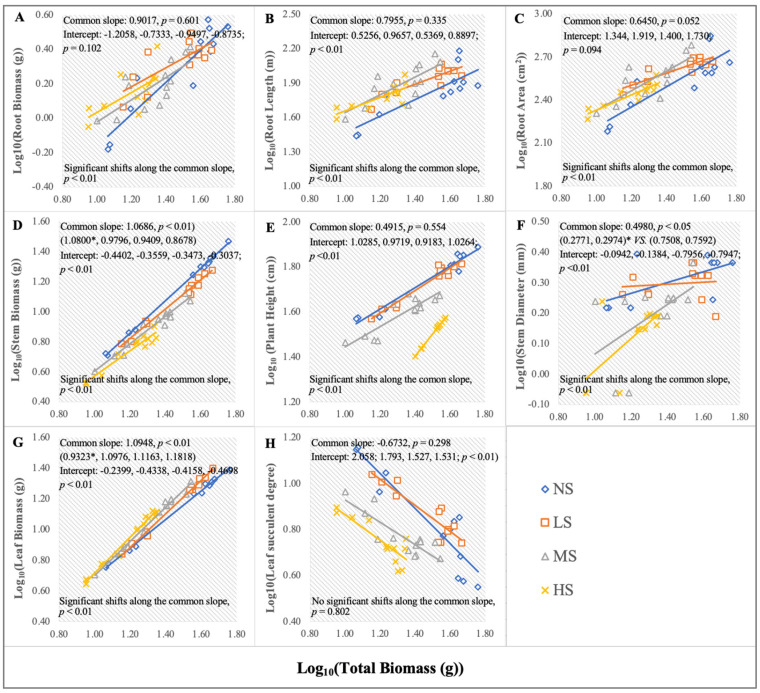
Allometric growth relationships between log total biomass and log root biomass (**A**), log RL (**B**), log RA (**C**), log stem biomass (**D**), log plant height (**E**), log stem diameter (**F**), log leaf biomass (**G**), and log leaf succulent degree (**H**) under the four NaCl levels throughout the sampling stages, as presented by Standardized Major Axis (SMA) regressions. The common slope test (*p* < 0.05) indicates significant differences among the groups and the group-caused differences are denoted with an asterisk. Significant elevation shifts and shifts along the common slope show that the relationship between the morphological components and total biomass allocation is affected by different NaCl levels.

**Figure 6 plants-13-01954-f006:**
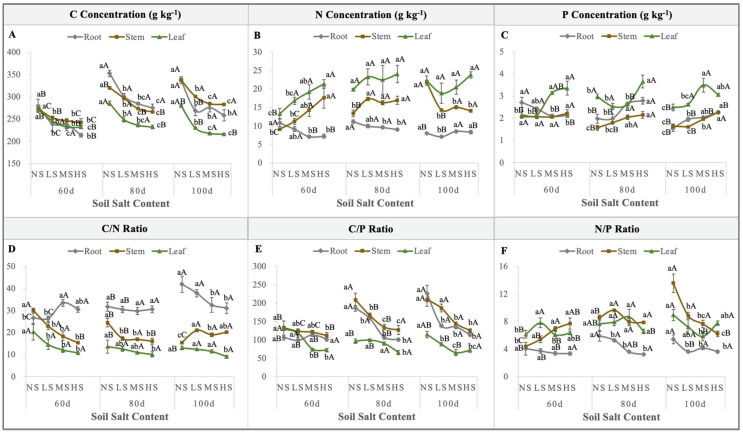
Variations in the C (**A**), N (**B**), and P (**C**) concentrations, and the C/N ratio (**D**), C/P ratio (**E**), and N/P ratio (**F**) of the roots, stems, and leaves under the four NaCl levels throughout the different sampling times. (Different lowercase letters indicate significant differences among the treatments (*p* < 0.05); different uppercase letters indicate significant differences among the sampling times (*p* < 0.05)).

**Figure 7 plants-13-01954-f007:**
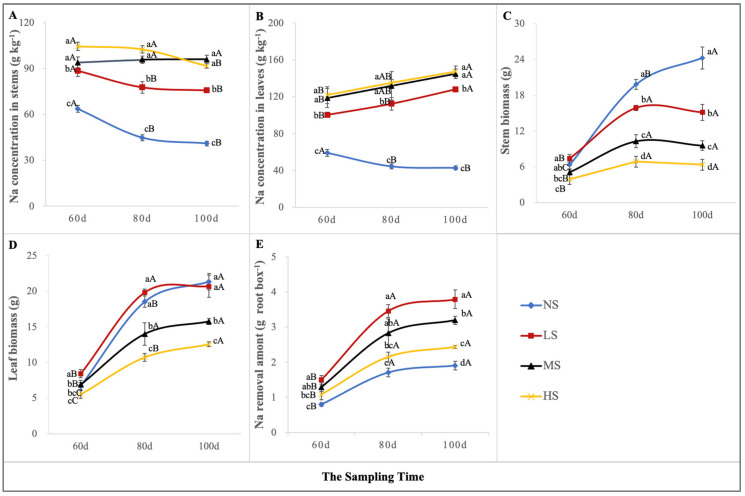
The concentration of Na in stems (**A**) and leaves (**B**), stem biomass (**C**), leaf biomass (**D**), and Na removal amount (**E**) under the four NaCl levels. (Different lowercase letters indicate significant differences among the treatments (*p* < 0.05); different uppercase letters indicate significant differences among the sampling times (*p* < 0.05)).

**Figure 8 plants-13-01954-f008:**
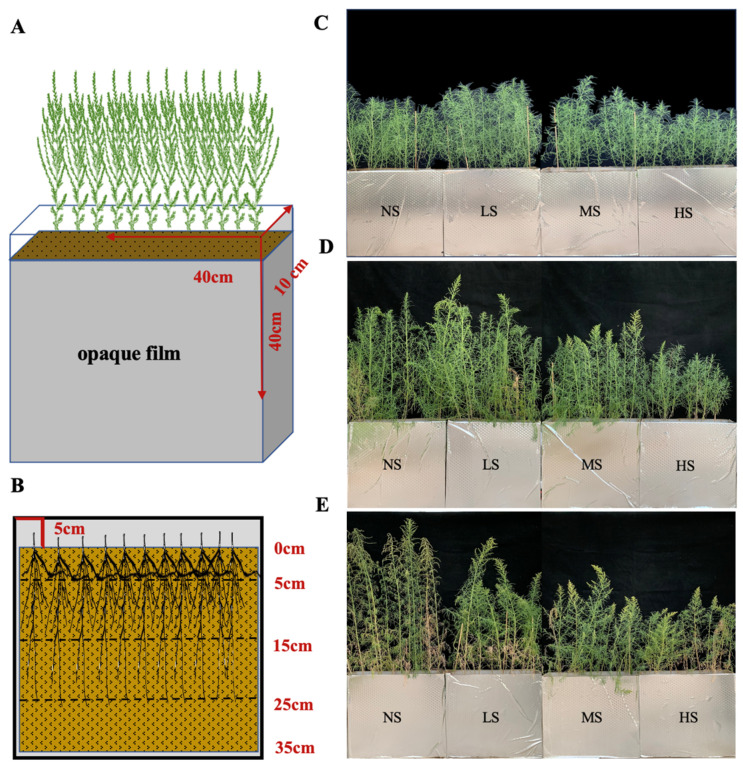
Root box design (**A**), below-ground sampling diagram (**B**), and above-ground apparent character of *S. salsa* under the four NaCl levels on the 60th day (**C**), 80th day (**D**), and 100th day (**E**).

**Table 1 plants-13-01954-t001:** Initial soil physical and chemical properties.

Soil Depth (cm)	pH	EC	Total Soluble Salt	Bulk Density	Field Water Capacity	Organic Matter	Available N	Olsen-P	Available K
		(mS cm^−1^)	(g kg^−1^)	(g cm^−3^)	(%)	(g kg^−1^)	(mg kg^−1^)	(mg kg^−1^)	(mg kg^−1^)
0~20	8.26	1.27	2.20	1.32	26.50	11.80	18.54	23.64	560.92

## Data Availability

Data are contained within the article.

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
