# Peer review of "The Relationship between Allometric Growth and the Stoichiometric Characteristics of Euhalophyte Suaeda salsa L. Grown in Saline–Alkali Lands: Biological Desalination Potential Prediction"

_plants, 2024, doi:10.3390/plants13141954_

Round 1

Reviewer 1 Report

Comments and Suggestions for Authors

The paper is very interesting and clearly written except the beginning of results (see below). I have no major comments and congratulate the authors for their paper. The only concern I had was the position of the materials and methods after the results and discussion which is weird for scientific papers but maybe the case in Plants?

-          Line 16, abstract : each acronyme should be fully written in brackets in the abstract.

-          Lines 36-39 : first sentence needs a reference.

-          Line 68 : use plural : euhalophytes target.

-          Line 106 : period instead of comma.

-          Line 107 : why period before and ? This paragraph is weirdly written.

-          Line 124 : remove And. You cannot begin a sentence by And.

Comments on the Quality of English Language

Comments on quality of the english are provided above.

Reviewer 2 Report

Comments and Suggestions for Authors

This manuscript aims at describing the allometric adjustments that a halophyte plant (Suaeda Salsa) shows when grown under different salinity conditions. Morphological parameters and concentration of different elements were registered during growth of the plants. The objective was to characterize the morphological plasticity under different salt regimes and to estimate the desalting potential of the plant.

While the experimental design and comparison between treatments are correct, there is a major flaw in the assumptions made for the biological desalinization potential prediction. Authors cite Kleber’s law but:

a. The name is incorrect. It's Kleiber

b. The Kleiber relation is for metabolism, not elemental composition

c. The manuscript referenced is for animal body size and metabolic rate (mammals, homeotherms), not plants

Therefore the section of the manuscript should be completely rewritten

Minor issues

1. Please define initials the first time that appear in the text.

2. Abstract

Please correct the expression of the biomass to 48.65g/root box or  48.65g root box-1 with the number -1 as superindex

3. Line 50

Reference 7 is incorrect

4. Line 153

The shape of the curves shoud be described in a more scientific way

5. Line 170

Reference to Figure is incorrect.

6. Line 174

The intercept data is not shown. It would be interesting to expose the values.

7. Line 186

The slope is negative in the figure 5H, but it does not decrease. The succulent degree decreases with the increase in plant size. Please correct

8. Line 192

With the data shown up to this part of the manuscript, it is not possible to make such affirmation

9. Line 202

Again, it would be interesting to show data

10. Line 237

“…we found N preferred to increase leaf biomass, followed by stem biomass (Figure 6B).”

This is not correct. Figure 6B shows that N concentration is higher in leaves and stems than in roots

Comments on the Quality of English Language

There are some grammatical mistakes in a number of sentences. 

Although the manuscript is generally well understood, reading would be more fluid if a native speaker reviewed the manuscript.

Reviewer 3 Report

Comments and Suggestions for Authors

The paper is basically descriptive and reports absolutely preliminary data. No mechanistic aspects supporting the results obtained as a result of salinity stress are explored in depth. Therefore, it is my opinion that it cannot be accepted for publication.

Comments on the Quality of English Language

There are some mistakes in English, so careful linguistic revision is needed

Author Response

This manuscript is only a study of plant ecology and does not involve in advances in plant physiology.

Round 2

Reviewer 2 Report

Comments and Suggestions for Authors

All questions have been answered. The authors have followed all the recomendations and now the manuscript is suitable for publication

Comments on the Quality of English Language

Correction of minor grammar mistakes could improve the quality of the manuscript

Author Response

The minor grammar mistakes have been revised. 

Reviewer 3 Report

Comments and Suggestions for Authors

The work is considerably improved compared with the previous version, although explanations and mechanistic aspects are still not well discussed and highlighted.

I recommend that the authors rewrite the introduction in a clearer and more logical form. 

As for the results part I suggest condensing it and avoiding sentences that are more suitable for discussion.

In the discussion, I would avoid repeating the results and better discuss the observed responses in relation to the salt stress conditions investigated in that study. 

Comments on the Quality of English Language

English language should be revised
